# Towards an Autonomous Industry 4.0 Warehouse: A UAV and Blockchain-Based System for Inventory and Traceability Applications in Big Data-Driven Supply Chain Management [note 1]

**DOI:** 10.3390/s19102394

**Published:** 2019-05-25

**Authors:** Tiago M. Fernández-Caramés, Oscar Blanco-Novoa, Iván Froiz-Míguez, Paula Fraga-Lamas

**Affiliations:** Department of Computer Engineering, Faculty of Computer Science, Universidade da Coruña, 15071 A Coruña, Spain; o.blanco@udc.es (O.B.-N.); ivan.froiz@udc.es (I.F.-M.)

**Keywords:** UAV, drones, Industry 4.0, traceability, blockchain, inventory, supply chain management, logistics, RFID, smart contracts, DLT, IPFS

## Abstract

Industry 4.0 has paved the way for a world where smart factories will automate and upgrade many processes through the use of some of the latest emerging technologies. One of such technologies is Unmanned Aerial Vehicles (UAVs), which have evolved a great deal in the last years in terms of technology (e.g., control units, sensors, UAV frames) and have significantly reduced their cost. UAVs can help industry in automatable and tedious tasks, like the ones performed on a regular basis for determining the inventory and for preserving item traceability. In such tasks, especially when it comes from untrusted third parties, it is essential to determine whether the collected information is valid or true. Likewise, ensuring data trustworthiness is a key issue in order to leverage Big Data analytics to supply chain efficiency and effectiveness. In such a case, blockchain, another Industry 4.0 technology that has become very popular in other fields like finance, has the potential to provide a higher level of transparency, security, trust and efficiency in the supply chain and enable the use of smart contracts. Thus, in this paper, we present the design and evaluation of a UAV-based system aimed at automating inventory tasks and keeping the traceability of industrial items attached to Radio-Frequency IDentification (RFID) tags. To confront current shortcomings, such a system is developed under a versatile, modular and scalable architecture aimed to reinforce cyber security and decentralization while fostering external audits and big data analytics. Therefore, the system uses a blockchain and a distributed ledger to store certain inventory data collected by UAVs, validate them, ensure their trustworthiness and make them available to the interested parties. In order to show the performance of the proposed system, different tests were performed in a real industrial warehouse, concluding that the system is able to obtain the inventory data really fast in comparison to traditional manual tasks, while being also able to estimate the position of the items when hovering over them thanks to their tag’s signal strength. In addition, the performance of the proposed blockchain-based architecture was evaluated in different scenarios.

## 1. Introduction

The concept Industry 4.0 fosters the evolution of traditional factories towards smart factories through the use of some of the latest advances in paradigms and technological enablers like big data [1], augmented reality [2,3,4,5], robotics [6], cyber–physical systems [7], fog computing [8,9] or the Industrial Internet-of-Things (IIoT) [10]. For instance, the application of advanced Big Data analytics in supply chain management can help to improve decision making for all activities across the supply chain. In particular, in inventory tasks [11], big data can help organizations to design modern inventory optimization systems, predict inventory needs, respond to changing customer demands, reduce inventory costs, obtain a holistic view of inventory levels, optimize the flow and storage of inventory, or even reduce safety stock.

Unmanned Aerial Vehicles (UAVs) are also considered a key technology for smart factories, since they allow carrying out repetitive and dangerous tasks without almost any human intervention or supervision. In the last years, UAVs proved to be really useful in fields like remote sensing (e.g., mining), real-time monitoring, disaster management, border and crowd surveillance, military applications, delivery of goods, precision agriculture, infrastructure inspection or media and entertainment, among others [12,13]. In many of such fields, UAVs perform tasks that constitute one of the foundations of Industry 4.0: to collect as much data as possible from multiple locations dynamically. In addition, UAVs not only collect data, but are also able to store, process and exchange information with suppliers or with devices deployed in factories.

Industry 4.0 technologies have to be integrated horizontally so that manufacturers and suppliers can cooperate. In order for a company to dynamically determine its need for supplies, it is necessary to keep track of their stock. For such a purpose, many companies carry out a periodic inventory and decide whether more supplies have to be purchased. Unfortunately, in many companies, such an inventory is performed manually, making it a really costly, time-consuming and tedious task. There exists software to automate stock control, but when it is operated by humans, the process is prone to accounting errors and it is usually not carried out in real time. Therefore, the ideal inventory should be performed automatically, in real-time and in an efficient, flexible and safe way.

UAVs have been applied to inventory and traceability tasks in the past. For instance, some of the latest commercial UAV-based systems [14,15,16] make use of a scanner installed on a UAV that performs a predefined flight in order to read barcodes. In the literature, there are more ambitious solutions like the one presented in [17], which describes an autonomous UAV that makes use of Radio-Frequency IDentification (RFID) and self-positioning/mapping techniques based on a 3D Light Detection and Ranging (LIDAR) device.

Blockchain and other Distributed Ledger Technologies (DLTs) are also predicted to be essential for many industries [18,19], since they allow for storing the collected data (or a proof of such data) so that they can be exchanged in a secure way among entities that do not trust each other. Although blockchain can be considered to be still under development in many aspects [20,21], some of its applications for fields where trust is a required (e.g., finance) have been already deployed [22]. Moreover, blockchain and other DLT enable the creation of smart contracts, which can be defined as self-sufficient decentralized codes that are executed autonomously when certain conditions of a business process are met. Thus, the code of a smart contract translates into legal terms the control over physical or digital objects through an executable program. For instance, a smart contract may be used as a sort of communication mechanism with a supplier when certain materials run low and one expects more incoming work that would require them.

Specifically, this article includes the following contributions:This is one of the first articles where design and testing of an RFID-based drone for performing inventory tasks are described. In fact, we have not found in the literature any other solution that makes use of a similar identification technology.Besides recent literature on blockchain-based autonomous business activity for UAVs [23], to our knowledge, this article is the first that proposes the theoretical design and practical implementation of a UAV-based, versatile, modular and scalable architecture aimed at fostering cyber security (specially, data integrity and redundancy) by including together the use of blockchain and a decentralized database solution. Specifically, the proposed system can use a blockchain to receive the inventory data collected by UAVs, validate them, ensure their trustworthiness and make them available to the interested parties. Moreover, the system is able to use smart contracts to automate certain processes without human intervention.We also evaluated the performance of the proposed decentralized database and the implemented smart contracts in diverse scenarios.

The rest of this paper is structured as follows. Section 2 reviews the previous work related to the use of UAVs in industrial applications, as well as the state-of-the-art of technologies for industrial inventory and traceability. Section 3 details the architecture of the proposed system, while Section 4 describes its implementation. Finally, Section 5 is devoted to the experiments and Section 6 to the conclusions.

## 2. Related Work

The solution proposed and evaluated in this article requires four essential elements to keep product traceability and obtain the inventory of a warehouse:A labelling technology: items need to be attached to tags or labels that are associated with a unique identifier and, in some cases, with additional information on the item.An identification technology: the labels or tags attached to the items have to be read remotely to automate the inventory processes.A UAV: although most labelling technologies make use of handheld readers, this article proposes the use of a UAV to automate and to accelerate data gathering.Supply management techniques: data gathering, processing and storing processes need to be efficient when handling a relevant number of information.

The most relevant aspects of these four elements are first studied and then analyzed in the following sub-sections in order to later determine the main components of the designed system.

### 2.1. Labelling and Identification Technologies

Currently, one of the most popular identification technologies is barcodes, which are essentially a visual representation of Global Trade Item Number (GTIN) codes [24]. However, barcode readers need Line-of-Sight (LoS) with the barcode label to read it correctly and it can only be read at relatively short distances (just a few tens of centimeters). Despite the mentioned limitations, barcodes improved inventory speed in industrial scenarios with respect to traditional manual identification procedures. In addition, barcode label manufacturing cost is really low, since they only require specific software and a printer.

In the middle of the 1990s, barcodes evolved towards bidimensional codes known as BiDimensional (BiDi) or Quick Response (QR) codes, which are able to store data (usually more than 1800 characters) and can be read with a simple smartphone camera, thus, reducing the overall cost of the inventory/traceability system. Nonetheless, QR codes can only be read up to a distance that depends on the size of the QR marker (the reading distance is often estimated to be less than ten times of the QR code diagonal).

Since both barcodes and QR codes are limited by their reading distance and their need for LoS between the reader and the code, they evolved towards more sophisticated labels (Figure 1 illustrates such an evolution). Labels based on RFID are considered the next step in the inventory and traceability system evolution [25]. Such a kind of label can be read at a distance that goes from several centimeters to several tens of meters [26]. In fact, reading distance is commonly related to the type of RFID tag: passive tags (i.e., tags that do not depend on batteries for carrying out RFID communications) reading distance usually does not exceed 20 m, while active tag communications (i.e., the ones that do use batteries for carrying out RFID communications) can easily reach 100 m in unobstructed environments. In addition, certain RFID tags can store information, which may be useful in certain inventory and traceability processes. It must be also noted that, in the last years, academia and industry have evolved RFID tags in order to add to them sensing capabilities, thus, creating tags that are able to measure temperature [27], acceleration [28] or light [29].

The next step in the tag evolution are the so-called smart labels [30], which are noticeably more complex than RFID tags and allow for providing industry 4.0 functionality like event detection [31], operator interactivity, a display for visual feedback, one or more communication technologies, positioning services or embed IoT sensors that collect environmental data that are relevant for the state of a product. Omni-ID’s View smart labels [32] are one of the few on the market and embed an e-ink display, flash memory storage, active and passive UHF RFID transceivers, and an infrared receiver for beacon-based positioning.

Besides the identification technologies related to the previously mentioned labelling solutions, there are others that have been suggested. For instance, Near Field Communication (NFC) [33] has been used in inventory applications, but it is only dedicated to scenarios where a short reading distance (often less than 30 cm) is needed.

There are other technologies, which, have been actually conceived as communications technologies, but can also provide identification mechanisms. For instance, Bluetooth Low Energy (BLE) is a generic Wireless Personal Area Network (WPAN) technology that usually does not reach more than 10 m when used through smartphones or up to 100 m in industrial applications [34]. There are diverse BLE profiles and specific devices called beacons that can be useful in certain inventory and traceability applications, as well as in indoor locations [35,36] and IoT applications [37,38], since they transmit a periodic signal to be located and identified.

The Media Access Control (MAC) address of a WiFi device (i.e., a device compatible with the IEEE 802.11 family of standards, including its vehicular version [39,40]) can also be used for identification purposes, as well as other technologies less frequently used in inventory tags, like devices based on ultrasounds, infrared communications, ZigBee [41], Low-Power Wide-Area Network (LoRA) [42], Dash7 [43], Ultra Wide Band (UWB), WirelessHART [44], RuBee (IEEE standard 1902.1), SigFox [45], ANT+ [46] or IEEE 802.11ah, among others.

As a summary, the main characteristics of the most relevant identification technologies for inventory and traceability applications are shown in Table 1, which specifies their frequency band, usual maximum range, data rate, power consumption, relevant features and some examples of applications that have made use of each technology.

### 2.2. UAVs for Inventory and Traceability Applications

UAVs have been suggested for providing services in a number of industrial applications, like for critical infrastructure inspections [47,48,49,50,51] or sensor monitoring [52,53,54]. It is important to note that industrial environments require certain UAV features that may differ from other applications, like obstacle collision avoidance [55], automatic cargo transport [56,57] or logistics optimization for swarms of UAVs [58].

UAVs have already been used for traceability and inventory management applications. For example, in [59], the authors use a QR code-based UAV to detect items. Although the system shows an impressive precision of 98.08%, it requires LoS with QR codes, which in practice limits its application to certain scenarios. Another interesting article is [60], where UWB devices are used to accurately position (i.e., with an indoor location accuracy of only 5 cm) a UAV that performs inventory management tasks.

RFID is the identification technology embedded in a UAV in [61]. In such a paper, the scenario is an open storage yard that is monitored by a DJI Phantom 2 drone that carries a passive UHF reader embedded by a PDA. Researchers from MIT have also conceived a similar theoretical system where multiple UAVs carry RFID readers to aid inventory automation in a warehouse [62]. Finally, it is worth mentioning the work in [63], which proposes the collaboration of an Unmanned Ground Vehicle (UGV) and a UAV to perform inventory tasks in a warehouse. Specifically, the UGV is used as a ground reference for the indoor flight of the UAV, which embeds a camera that recognizes augmented reality markers. Thus, the UGV carries the UAV to places where there are items to be inventoried and then the UAV flies vertically to scan them, sending the collected data to a ground station.

### 2.3. Big Data for Inventory and Supply Chain Management

Big Data analytics can also be applied to optimize knowledge extraction and decision-making in inventory and supply management applications. A comprehensive review on the application of big data to such fields can be found in [64], with the management of safety stocks being one of the key aspects [65]. Moreover, other authors have already analyzed how to face the current challenges and opportunities that the transformation of supply chain management suppose in order to be driven by big data techniques [66].

With respect to practical deployments, in [67], the authors propose a big data analytics framework that processes the information collected from an RFID-enabled shop floor. The authors define, at a logistics management level, different Key Performance Indicators (KPIs) in order to evaluate different manufacturing objects and the main findings are converted into managerial guidance for decision making.

Regarding the use of big data analytics for inventory applications, several advantages can be pointed out [11]: it can help to take informed decisions related to inventory performance, it may aid to obtain a holistic view at inventory levels across the supply chain and with external stakeholders, and it enables to predict inventory needs and changes in customer demand using statistical forecasting techniques, as well as to reduce inventory costs dramatically.

### 2.4. Analysis of the State of The Art

Table 2 shows a comparison of the most relevant characteristics of the previously cited UAV-based inventory systems together with the ones of the system proposed in this article. As it can be observed after reviewing the different aspects of the state-of-the-art, it is possible to highlight several important shortcomings that motivated this article.

First of all, few papers detail practical UAV-based applications for inventory and traceability applications, specially with RFID as the identification technology embedded in the UAV. Specifically, most of them do not take into consideration in the UAV design a trade-off between cost, modularity, payload capacity and robustness. Second, the emergence of Big Data analytics impose additional issues to emerging solutions that are still open for research like:Data volume. The amount of heterogeneous data generated by Industry 4.0 technologies has increased exponentially, especially the information collected from Industrial Internet of Things (IIoT) devices and remote entities. These data have to be hosted, distributed and computed across a number of organizations ensuring a certain degree of operational efficiency.Speed. Decisions should be made as quickly as possible (ideally, in real time), so there is a need for speeding up the processes related to them: data production, data collection, reliability, data transfer speed, data storage efficiency, knowledge extraction and analysis, as well as decision-making.Verification and veracity. There is a myriad of factors that may derive into collecting bad data (e.g., noise, inaccurate readings), so they should be verified so that only valid or true data are further processed.Versioning. Massive datasets should be linked and the accidental disappearance of important data should be prevented.Accessibility. Stakeholders must be able to access data through resilient networks, which enable persistent availability with or without Internet connectivity.

Although different alternatives have been proposed for facing some of the issues mentioned above, DLTs seem to be the most promising solution in order to guarantee operational efficiency and data transparency, authenticity and security. However, there is a shortcoming related to the fact that, although there are examples in the literature of DLTs applied to other sectors, to the knowledge of the authors, there are no examples of DLTs or blockchain-based architectures for the upcoming big data-driven supply chain applications.

Finally, it must be noted that the use of current DLTs, and specifically blockchain, involve certain weaknesses related to the immature status of the technology, like the lack of scalability, high energy consumption, low performance, interoperability risks or privacy issues [18,19,20,21,22]. Moreover, blockchain technologies face design limitations in transaction capacity (i.e., throughput and latency), in validation protocols or in the implementation of smart contracts. Therefore, architecture design should consider these constraints together with the specific decentralization requirements. Furthermore, as of writing, there are several open issues that require more research. For instance, no references were found in the literature on the evaluation of the performance of blockchain-based application architectures.

As a consequence of the previous analysis, the design of the system proposed in this article has been devised taking the previous shortcomings into consideration.

## 3. Design of the System

Figure 2 depicts the proposed communications architecture. In such an architecture, a UAV carries a Single-Board Computer (SBC) and a tag reader. The tag reader is used for collecting data from wireless tags that are attached to the items to be inventoried or traced. Regarding the SBC, it collects tag data from the tag reader, processes them and sends them through a wireless communications interface to a ground station. Then, to provide enhanced cybersecurity, the ground station can make use of its internal software modules to send the collected information to two possible destinations: to a decentralized remote storage network or to a blockchain.

In the case of sending the data to a blockchain, the ground station makes use of a software module that acts as a blockchain client. Therefore, the SBC is able to store in a secure way the collected data (or their hashes) into the remote blockchain, which also allows the proposed system to participate in smart contracts that enables the direct participation of suppliers, manufacturers, retailers or logistics operators [68]. It is important to note that very similar features may be provided by traditionally centralized solutions like databases, cloud-based services or complex-event processing software, but blockchain technologies include the following features that make them really attractive for industrial scenarios where third parties may be interested in carrying out audits or financial evaluations, or may need to verify the fulfillment of certain laws [21]:Transparency. Blockchain and smart contracts allow for providing access to inventory and traceability information to third-parties, which can monitor it and determine whether it has been tampered with.Application decentralization. Medium and large software deployments usually require to make use of centralized servers that are often expensive to deploy and maintain [69,70]. Blockchain is one of the technologies that enable application decentralization and, at the same time, avoids the involvement of middlemen that provide outsourced centralized solutions.Data authenticity. In many industries, it is essential to trust the inventory and traceability data received from suppliers, manufacturers or governments. Blockchain enables to implement mechanisms that provide accountability and, as a consequence, trustworthiness. In addition, it is worth noting that such data trustworthiness is essential for the effectiveness of the application of Big Data techniques.Data security. Blockchain allows for preserving the privacy and anonymity of the data exchanged with other entities, so that they remain private to non-authorized parties.Operational efficiency. Blockchain technologies are able to automate the verification of the attributes of a transaction in an inexpensive way.

With respect to the decentralized storage, it is included in the architecture in order to provide the following three main advantages that useful for enhancing the security of the inventory and traceability data:Redundancy. Decentralized storage systems, when properly synchronized, are able to create copies of the stored data automatically in multiple nodes, so the information is not available from a single source that may constitute a point of failure.Tolerance to cyber-attacks. Since the information can be available from multiple data storage nodes, if one or several of them are taken down by cyber-attacks (e.g., Denial of Service (DoS) or Distributed Denial of Service (DDoS) attacks), it may be possible to access it through the other nodes.Ability to run decentralized applications (dApps). Besides pure file storage, it is possible to develop and deploy dApps that provide the previously mentioned features (i.e., redundancy and increased security). In addition, it is possible to develop dApps that run on decentralized storage systems while cooperating with a blockchain [71].

Both the blockchain and the decentralized storage system essentially manage the same data types as in other traditional inventory solutions: sets of unique alphanumeric identifiers (UIDs). Thus, item UID association and information processing (e.g., to determine the number of items available of one specific type) need to be performed by an additional software layer that may be implemented through a Cyber-Physical System (CPS). Moreover, note that, although remote users may access the collected raw information directly from one of the nodes of the decentralized storage network or through a blockchain browser, it is usually more practical to make use of a user-friendly interface like the one provided by the CPS. Thus, a CPS collects and processes the data needed by remote users and presents them in a way that can be easily understood by remote users [72].

## 4. Implementation

### 4.1. UAV Implementation

UAVs vary widely in size, materials, components and configuration. In the design of the proposed UAV, the main objective was to develop a cost-effective, simple and modular initial prototype that can be easily adapted to different applications, scenarios and/or performance criteria.

Figure 3 depicts the main components of the designed UAV, while Table 3 shows a summary of their characteristics. A multipurpose UAV was designed with both indoor and outdoor flight capabilities in order to extend its use to different applications and maximize its exploitation opportunities. The chosen configuration offers a good trade-off between cost, payload capacity and reliability; an hexacopter configuration has the minimum number of motors in order to be able to recover and land in the case of a motor failure without damaging people or goods. More rotors would increase the reliability even further but would also increase the cost and its total weight.

Specifically, the UAV is mounted on an hexacopter frame of 550 mm of diameter mostly made out of carbon fiber except for the arms, which are made of plastic reinforced by carbon fiber rods in the interior. The flight controller is a PixHawk 2.4.8 that has been flashed with the well-known open-source firmware Ardupilot [73] The thrust to move the UAV is generated by six 920 Kv brushless motors controlled by six 30 A Electronic Speed Control (ESCs), which are powered by a four-cell 5 Ah Li-Po battery that also provides power to all the on-board electronics through a voltage conversion module. Besides the built-in sensors of the flight controller board, a UBLOX M8N GPS module was included to enable outdoor autonomous flights since in this industrial use case, there are storage areas outdoors in a nearby dock next to the warehouse where inventory can also be performed.

### 4.2. Implemented Architecture

Figure 4 shows the actual implemented architecture of the proposed system. It can be observed that it is very similar to Figure 2, but, since this article is focused on assessing the performance of the UAV-based inventory system, it includes certain simplifications on aspects that are not evaluated (i.e., on the CPS).

As it can be observed in Figure 3 and Figure 4, in order to perform the inventory, an RFID reader (NPR Active Track-2) is carried by the hexacopter. Specifically, active Ultra High Frequency (UHF) RFID has been selected as item identification technology due to its performance in environments with multiple metallic obstacles [26]. The RFID reader has been modified to reduce its weight by replacing its steel case with a lighter one made of foam, which protects the reader and reduces vibrations. In addition, it is worth pointing out that, as it can be observed in Figure 3, the RFID reader antennas were placed as far as possible from the hexacopter propellers, which can influence communications performance. Nonetheless, the location of the antennas may be further optimized.

Regarding the used RFID tags, they are active UHF tags from RF-Code [74] that can be read with the NPR Active Track 2 reader (e.g., Active Rugged Tag-175S). Such a reader is connected through an Ethernet cable to the SBC. The tag UIDs collected by the SBC are first stored locally in a JavaScript Object Notation (JSON) file and then sent through WiFi to the ground station server, which, for the experiments performed in this article, it was run in a laptop.

Figure 4 shows a high-level view of the implemented blockchain-based architecture. As it can be observed in the Figure, to persist the collected inventory data, the proposed system makes use of the decentralized database OrbitDB [75], which runs over InterPlanetary File System (IPFS) [76]. IPFS provides a high throughput content-addressed block storage model with content-addressed hyper links. This forms a generalized Merkle Directed Acyclic Graph (DAG), a data structure upon which one can build versioned file systems, blockchains, and even a Permanent Web. Specifically, the main features of IPFS are:It keeps every version of the files, making it simple to set up resilient networks.It removes duplication across the network, therefore saving in storage. Each network node stores only content it is interested in and some indexing information that helps to figure out who is storing what.Each file and all of the blocks within it are given a unique fingerprint (i.e., cryptographic hash).Every file can be found in an user-friendly way (e.g., by human-readable names) using a decentralized naming system called Inter-Planetary Name System (IPNS).It has no single point of failure and nodes do not need to trust each other.

OrbitDB is built on top of IPFS to create a serverless, distributed, peer-to-peer database in alpha-stage software. It uses IPFS as its data storage, as well as IPFS Pubsub to synchronize databases with peers automatically. In addition, OrbitDB provides various types of databases for different data models and use cases:log: an immutable (append-only) log with traceable history. Useful for “latest N” use cases or as a message queue.lfeed: a mutable log with traceable history. Entries can be added and removed.keyvalue: a key-value database.docs: a document database where JSON documents can be stored and indexed by a specified key. Useful for building search indices or version controlling documents and data.counter: Useful for counting events separate from log/feed data.

All the previously mentioned types of databases are implemented on top of ipfs-log, an immutable, operation-based Conflict-free Replicated Data Structure (CRDT) for distributed systems. If none of the OrbitDB database types match a dApp needs or if a specific functionality is needed, it allows for implementing a custom database easily.

With respect to the blockchain, two different Ethereum testnets (i.e., test networks that are isolated from the public Ethereum blockchain [77]) were used: Rinkeby [78] and Ropsten [79]. Rinkeby is a Proof-of-Authority (PoA) testnet created by the Ethereum team that makes use of the Clique PoA consensus protocol, where authorized signers are responsible for minting the blocks. In such a network blocks are created on average every 15 s and Ether cannot be mined (it is requested through a faucet [80]). In contrast, Ropsten is a Proof-of-Work (PoW) Ethereum testnet where Ether can be either mined or requested from a faucet [81]. Ropsten’s blocks are usually minted in less than 30 s and, although the testnet reproduces with more fidelity than Rinkeby Ethereum’s mainnet production environment, it is prone to Denial-of-Service (DoS) attacks (e.g., by increasing the block gas limit remarkably while sending large transactions through the network), which makes synchronization slow and makes clients consume a lot of disk space. Rinkeby and Ropsten testnets allow for executing smart contracts (compiled and deployed through Truffle [82]), which store in a string the JSON file with the inventory data and its hash, so the blockchain acts both as a immutable log and as a timestamping server.

In the implemented architecture, remote users are able to access the raw OrbitDB and Ethereum data, but, thanks to the proposed architecture, it is straightforward to develop a backend that collects data from OrbitDB and Ethereum, and presents them through a frontend.

### 4.3. Inventory Data Insertion and Reading Processes

Figure 5 illustrates all the components that are involved in the implementation of the part the architecture related to decentralized storage and the blockchain. Besides the previously mentioned components (i.e., OrbitDB, IPFS, the Ethereum testnets and Truffle), three another elements are used:Infura [83]: it provides an easy to use HTTP API for accessing Ethereum that can be even implemented by resource-constraint IoT devices.Node.js [84]: it is an open-source platform that allows for executing JavaScript code outside of a browser.Web3 Javascript API [85]: It is a collection of libraries that allow for interacting with Ethereum nodes. In the proposed architecture, the Web3 API is called from a Node.js instance that exchanges requests that are handled by Infura.

Figure 5 also shows the steps performed when inserting inventory data in OrbitDB and Ethereum. Before starting to store data, it is assumed that all the involved components are up and running, and that the different OrbitDB nodes synchronize their data periodically (this is illustrated by steps 0A and 0B, which indicate that every node is synchronized through a publish/subscribe scheme). Moreover, it is also assumed that the smart contracts haven been deployed through Truffle in one of the Ethereum’s testnets. Then, when a ground station wants to upload new inventory data, it first appends them to OrbitDB, which returns a hash as an acknowledgement of the transaction. Such an OrbitDB hash, together with the inventory data and the Infura authentication token, are sent to Infura through a data insertion request. If the authentication token is valid, Infura would take the inventory data and the OrbitDB hash, and would execute the setData function of the smart contract, so that it updates the stored inventory data values and the OrbitDB hash. As a result of this operation, Ethereum returns a hash that can be used later to request the blockchain transaction information.

Figure 6 illustrates the steps required by an entity that wants to verify that the data stored in OrbitDB have not being altered after their insertion. In such a case, the entity only would need to first request the inventory data from OrbitDB and them perform through Infura a *GetData* request to the corresponding Ethereum’s smart contract, which would return the inventory information stored on the blockchain and its OrbitDB hash. Then, the entity would only need to compare the inventory information collected from OrbitDB with the one from Ethereum, and determine whether it has been modified.

As it can be concluded from the proposed architecture and processes, they enable data trustworthiness through four different mechanisms:Information integrity can be verified by checking its hash. In addition, Ethereum and OrbitDB act as timestamping services, so it can be easily verified when the data were inserted.Since the data stored on the blockchain cannot be tampered without leaving a trace, the authenticity of the inventory information stored on OrbitDB can be easily checked.Every user transaction within OrbitDB is protected by asymmetric cryptography mechanisms that make use of a public and a private key.Similarly, the data that is exchanged with Ethereum are managed by Infura, which protects them through an API key and a secret key.

## 5. Experiments

### 5.1. Experimental Scenario

In order to evaluate the proposed system, it was tested in a big industrial warehouse that is shown in Figure 7. The warehouse is approximately 120 m long and 40 m wide, but, due to security reasons (the warehouse was in operation during the experiments), the tags were deployed in an isolated subarea of 50 m × 40 m.

As it can be observed in Figure 7, the warehouse stores items that are packaged into wooden, plastic or cardboard boxes. There are also items that are not packaged and, thus, only a plastic inventory label is attached to them. Due to such an item diversity, in order to provide a fair estimation of the performance of the proposed system, the deployed tags were attached to items made out of diverse materials, some of which are shown in Figure 8.

A total of 13 different tags were attached to items scattered throughout the previously mentioned isolated area of the warehouse that was monitored with the drone. Figure 9 shows the location of the ground station during the tests, while Figure 10 illustrates one of the moments during the experiments. As it can be observed in Figure 10 and in the video of the Appendix A, during the tests the drone was operated in manual mode in order to avoid possible physical security problems. Such an operation is expected to be automated thanks to the use of different on-board sensors and the placement of multiple image-based or RFID markers in the environment. In the same way, while during the tests the operator moved the drone to follow a circular path over the monitored area, in the future such a path will be prefixed by software through indoor waypoints.

### 5.2. Inventory Time

The firsts tests were carried out in order to determine how fast the inventory data could be collected. Figure 11 shows the percentage of read tags through time for four different tests. During such tests the tags remained at the same location and were attached to the same items. Considering the nature of the proposed experiments and the RFID reader reading range, the pilot experience and skills can be considered negligible in the results obtained.

An example of the collected data is shown in Table 4, which is related to Test 2 of Figure 11. Specifically, the table indicates the number and percentage of read inventory tags, and the time stamp collected by the drone every time a new tag is detected (its ID is indicated in the last column).

During Tests 1 and 2, the drone departed from the same spot and followed a similar path. This can be observed in Figure 11: the curves for both tests are very similar. Nonetheless, the inventory was gathered faster during Test 2: it only took approximately 78 s to collect the data, which is less than the 116 s needed during Test 1.

Although the tags were scattered randomly throughout the monitored area of the warehouse (with the only restriction of attaching them to items of the three most common materials), the departing point of the drone is essential for accelerating the inventory collection. This can be observed in the curves of Figure 11 related to Tests 3 and 4. In the case of Test 3 the departing point was located in the opposite side of the beginning of the monitored area, where several tags were placed. This derived into requiring a total of 229 s to collect all the UIDs of the inventoried items. However, in Test 4 the drone was roughly in the middle of the monitored area, thus accelerating remarkably their detection (it only took 26 s to complete the inventory).

Except for Test 3 (due to its specific location), in the other cases, mainly because of the reading range and the omni-directional antennas of the reader, during the first 11 s (as the drone rose from the ground), it was possible to read roughly 30% of the tags. Moreover, in all tests, 50% of the items where inventoried in less than a minute and all of them in less than four minutes. These results are really promising, since the time required by a human operator to collect the same information is far greater than when using the proposed UAV-based system (the operator needs to walk through the area, locate the items and identify them manually).

### 5.3. Signal Strength Monitoring

As the drone hovers above the warehouse, besides tag UIDs, it also collects the Signal Strength Indicator (SSI) of such tags, which can potentially be used for locating industrial items and for creating signal strength maps [86].

An example of the SSI fluctuation perceived by the drone for one of the RFID tags during an inventory flight is shown in Figure 12. It can be observed that while the drone is hovering around the warehouse, in locations far from the tag, the collected SSIs fluctuate between −50 and −62 dBm. However, once the drone is close to the tag, SSI levels go up to around −40 dBm and, as the drone moves away from the tag, the collected SSI values go down again to be between −50 and −62 dBm. Therefore, if the location of the drone can be obtained through an indoor positioning system, due to the existent correlation, tag locations may be estimated at the same time as the inventory data are collected. However, it must be noted that, in order to design an indoor positioning system for industrial environments, additional factors should be considered, like the reflections, diffraction and refraction caused by surrounding materials, the presence of hostile electromagnetic sources, certain features of the scenario (e.g., presence of metals, water, exposure to liquids, acids, salinity, fuel or other corrosive substances, tolerance to high temperatures) or the actual reading distance, among others [26].

Finally, it is worth pointing out that just after flying over the tag there were several seconds (roughly between seconds 120 and 160) during which the tag SSIs were not received by the UAV, what was probably caused by a signal blockage related to the presence of large metallic item in the scenario. This must be taken into account in future developments in order to determine the optimal tag and item positions to maximize SSI reception.

### 5.4. Performance of the Implemented Architecture

#### 5.4.1. Performance of the Decentralized Database

The performance of OrbitDB was measured in terms of response latency (i.e., how fast the inventory data were inserted into OrbitDB after being sent by a UAV ground station). Three scenarios (named A, B and C) were simulated in order to evaluate the effect of payload size and network delay, so small, medium and large payloads were sent to OrbitDB when it was running either in the same Intranet (i.e., the same private local network) as the ground station, and when it was running in a remote cloud on the Internet (i.e., for simulating the connection with other stakeholders such as suppliers or external audits). As a reference, it can be indicated that the minimum/average/maximum round-trip times to the machine that ran OrbitDB in the Intranet were 0.935/1.034/1.695 ms, while the same values for the connection between the ground station and the remote cloud were 37.948/39.362/51.172 ms. Scenarios A, B and C take place inside an Intranet. Scenario A corresponds exactly to the environment tested in Section 5.2: the inventory data were only 13 Tag IDs, which derived into a very small payload (around 1 KB). Scenarios B and C simulate inventory data of 5000 (around 30 KB) and 10,000 (around 67 KB) items, respectively. Regarding Scenarios D, E and F, they also simulate 13, 5000 and 10,000 items, respectively, but the inventory data is sent from the local network to a remote OrbitDB instance that runs on a cloud on the Internet.

These scenarios resulted into the six use cases characterized in Table 5, which shows, for each scenario, the number of inventoried tags, as well as the mean and variance of the obtained OrbitdB response latency. In addition, Figure 13 and Figure 14 show the response latency for the six evaluated scenarios and for 2000 insertion requests per scenario when measuring the time elapsed since each request is issued, until the OrbitDB hash is obtained (i.e., from step 1 to step 2 of Figure 5).

The results show that, in terms of response delay, OrbitDB insertions are really fast: in the worst tested case (Scenario F), the average inventory data insertion time requires roughly 0.55 s. In addition, it can be pointed out that, as expected, the larger the payload, the slower the insertion. However, the difference in response delay between the insertion of 13 and 10,000 tag IDs is negligible when OrbitDB runs on the Intranet (only 31.6 ms), while, in the same case but when OrbitDB was running of the remote cloud, the difference rose up to 190.5 ms. Moreover, at the view of the low variance values, it can be observed that response delay is very stable in both networks, although, obviously, it is clearly more stable for Scenarios A, B and C (i.e., on the Intranet).

Regarding Figure 13 and Figure 14, they seem to show a clear average response time, although such a time oscillates depending on different factors, like the network load (i.e., the network is actually shared with other users). Despite such oscillations, it is possible to obtain the Probability Density Function (PDF) of the different response delays. Figure 15 shows the PDFs for the three Intranet scenarios, which follow a Generalized Extreme Value (GEV) distribution with different values for the parameters of location (μ), scale (σ) and shape (k) [87]. In the case of the Internet scenarios, Figure 16 shows that they seem to follow a multimodal distribution (e.g., bimodal), although in the Figure it was fitted to a kernel distribution [88]. Therefore, the mentioned PDFs may be used in the future to model the behavior of the different scenarios and then generate artificial samples from the fitted distributions to perform Monte Carlo simulations to test the theoretical load that can be supported by the proposed system.

#### 5.4.2. Performance of The Blockchain

It is also interesting to measure the blockchain response latency in order to detect a possible bottleneck of the architecture. While Rinkeby is a PoA testnet whose time to mint a block is set to an average of 15 s, the Ropsten testnet is based on PoW and thus the same time may vary noticeably.

Figure 17 shows the response latency of almost 100 transactions (9.4 KB) in the Ropsten testnet during a smart contract update of the proposed system. Specifically, the elapsed time is measured since the transaction is included in a block until it is validated (from step 3 to step 5 in Figure 5). As it can be observed in Figure 17, the response time varies significantly from less to 5 s up to more than 70 s, being the delays significantly higher than in the OrbitDB performance tests.

Figure 18 shows a caption of Etherscan where the specific details of the transactions performed on the smart contract used for the tests performed in this Section (0x3A99AFac4A32b29C17 Aeb7d0B4E3C4F28EA200c7). The details of the transaction include the account balance, the performed transactions, the addresses of the involved parties and the number of miners. Additional details are available on https://ropsten.etherscan.io/address/0x3a99afac4a32b29c17aeb7d0b4e3c4f28ea200c7.

## 6. Conclusions

This article presented the design, implementation and evaluation of an UAV and blockchain-based system for Industry 4.0 inventory and traceability applications. After reviewing the most relevant initiatives related to industrial UAV applications and identification technologies, the architecture and components of the proposed UAV and RFID-based system were detailed. Such a system is able to collect and process inventory data in real-time and send them to a blockchain and to a decentralized storage network for providing enhanced cyber security, redundancy and the ability to run decentralized applications. Moreover, the system was able to use smart contract to automate certain processes without human intervention. The proposed system was tested in a real warehouse and the obtained results show that it is able to collect inventory data remarkably faster than a human operator and that it is possible to locate items in the warehouse by using their SSI. Furthermore, the performance of the proposed blockchain-based architecture was evaluated in diverse scenarios.

## Figures and Tables

**Figure 1 sensors-19-02394-f001:**
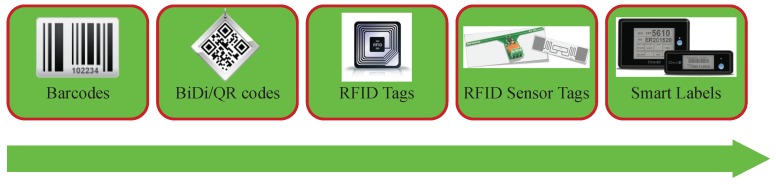
Technological evolution of identification tags.

**Figure 2 sensors-19-02394-f002:**
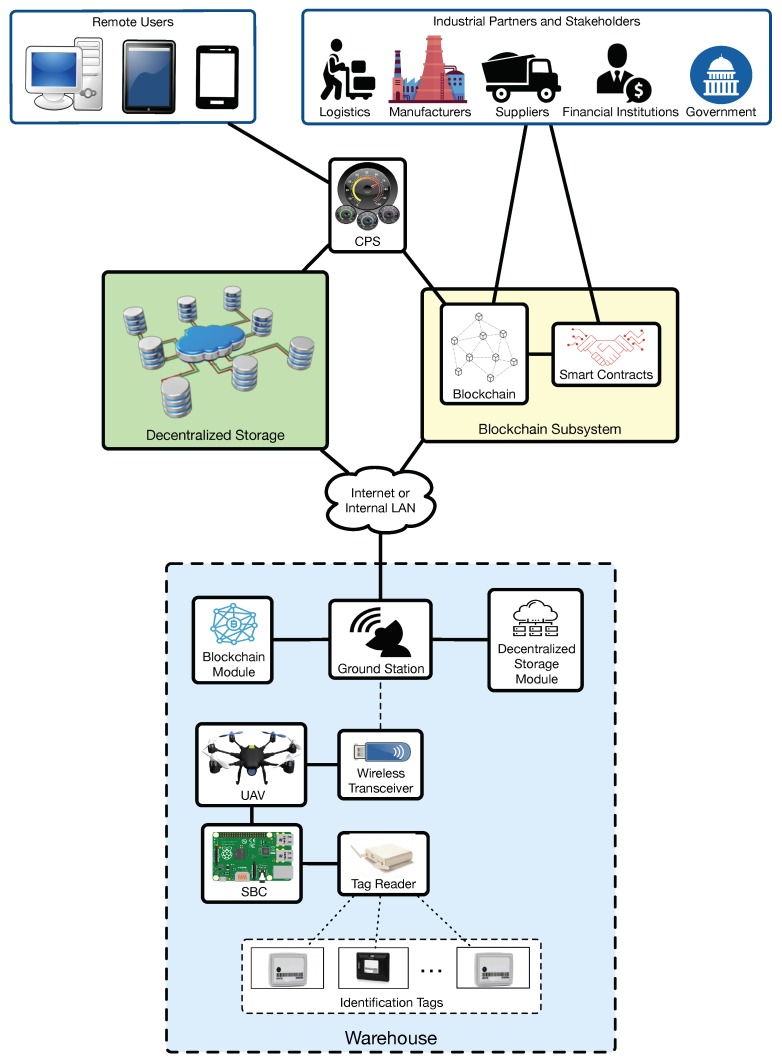
Proposed communications architecture.

**Figure 3 sensors-19-02394-f003:**
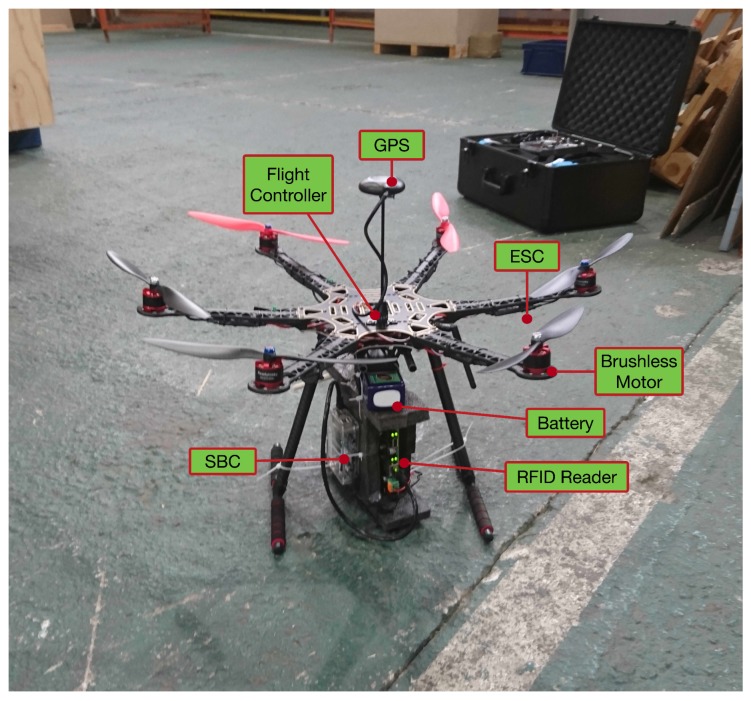
UAV used for the inventory and traceability system.

**Figure 4 sensors-19-02394-f004:**
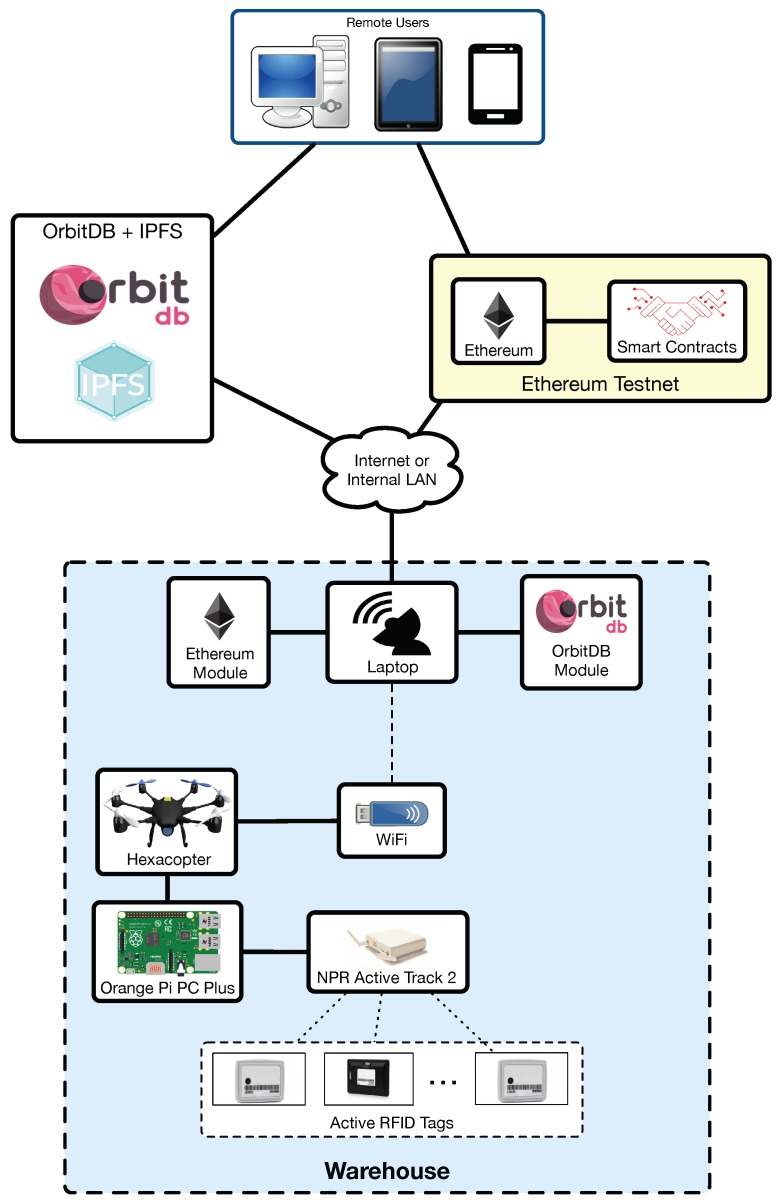
Implemented architecture.

**Figure 5 sensors-19-02394-f005:**
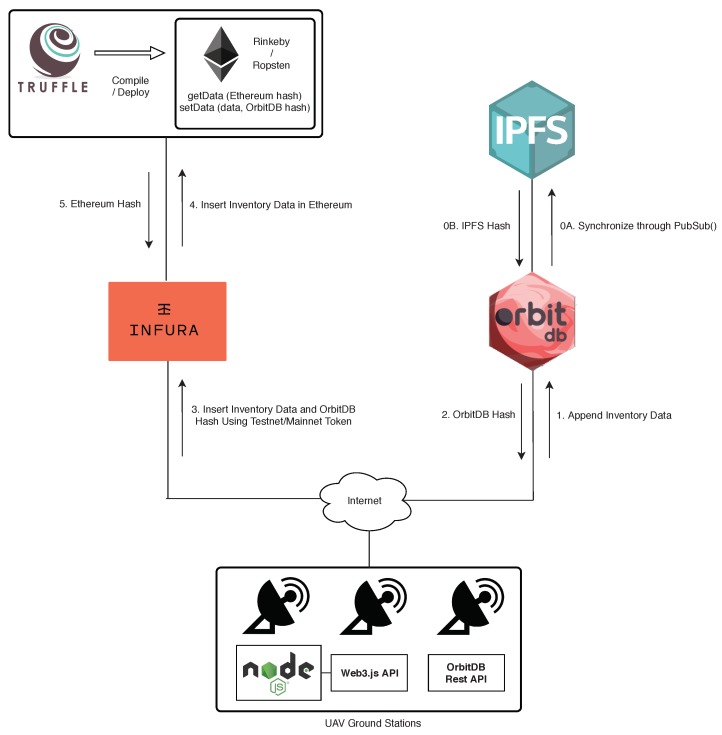
Inventory data insertion process and implemented architecture.

**Figure 6 sensors-19-02394-f006:**
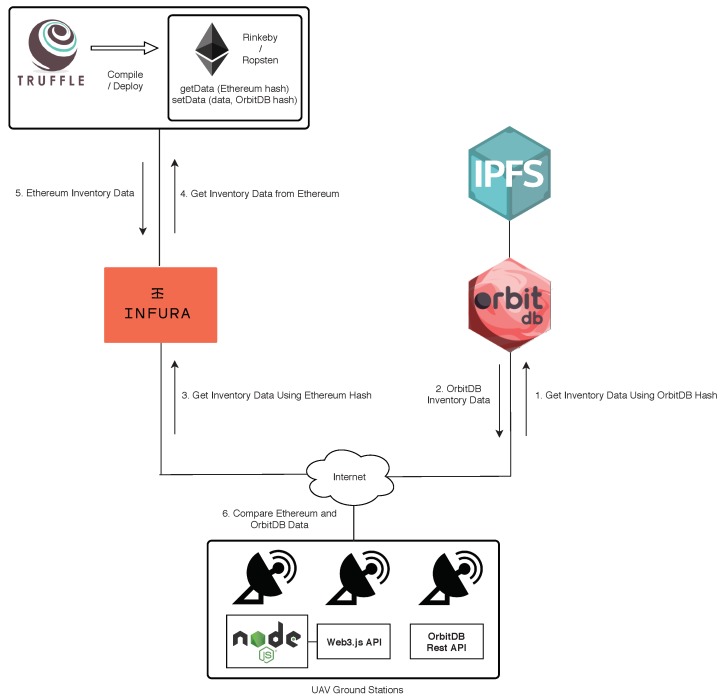
Inventory data verification process.

**Figure 7 sensors-19-02394-f007:**
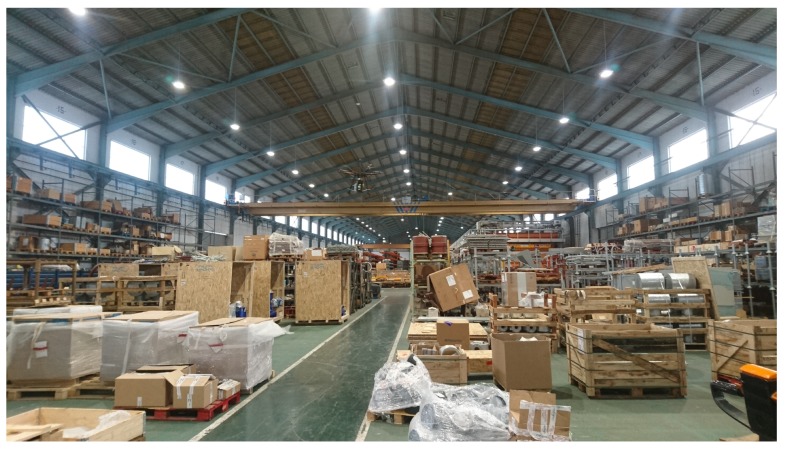
Warehouse where the experiments were performed.

**Figure 8 sensors-19-02394-f008:**
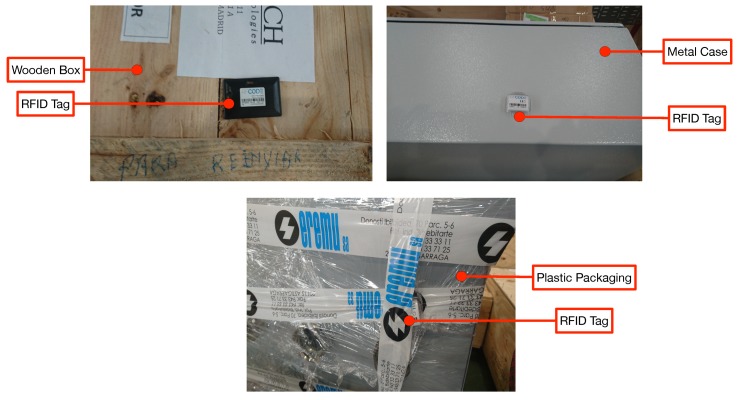
Warehouse material diversity.

**Figure 9 sensors-19-02394-f009:**
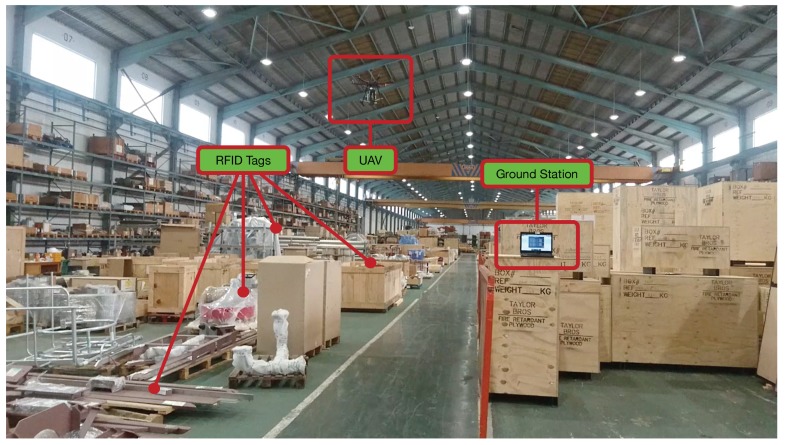
Ground station setup.

**Figure 10 sensors-19-02394-f010:**
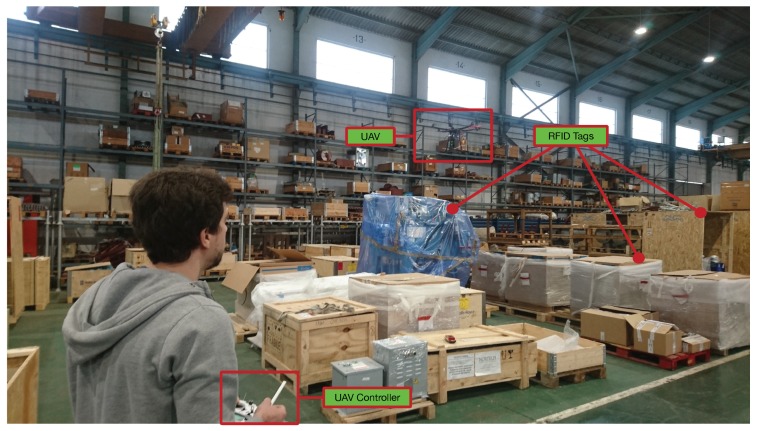
One of the instants during the inventory tests.

**Figure 11 sensors-19-02394-f011:**
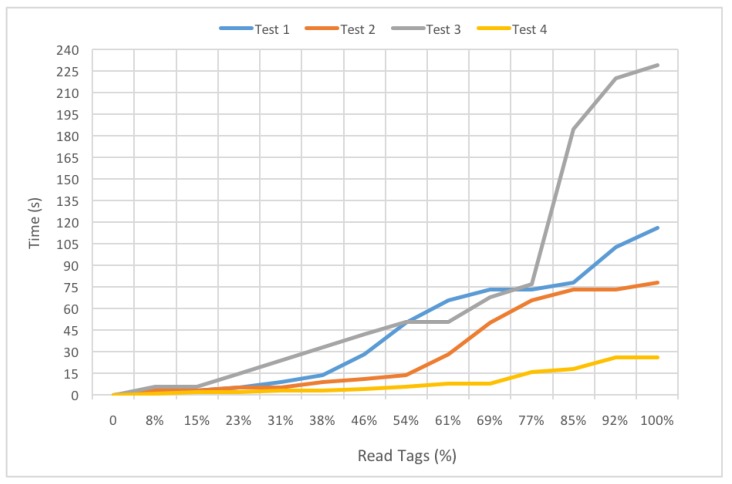
Percentage of read tags during four inventory flights.

**Figure 12 sensors-19-02394-f012:**
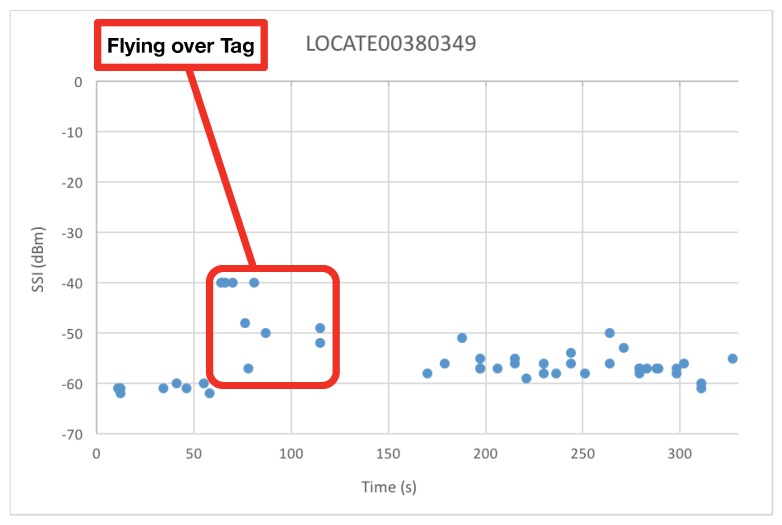
SSI evolution of a tag during an inventory flight.

**Figure 13 sensors-19-02394-f013:**
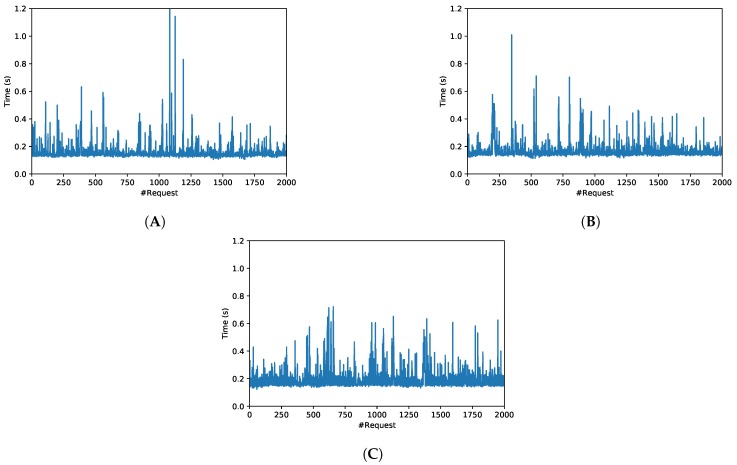
Response time in OrbitDB for (**A**) Scenario A, (**B**) Scenario B and (**C**) Scenario C.

**Figure 14 sensors-19-02394-f014:**
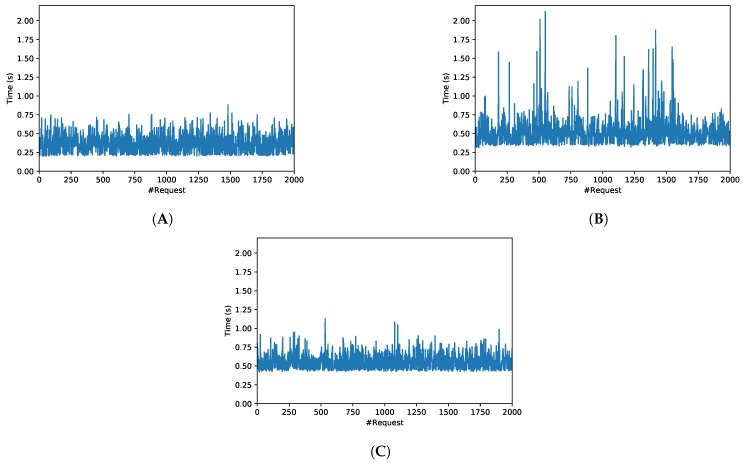
Response time in OrbitDB for (**A**) Scenario D, (**B**) Scenario E and (**C**) Scenario F.

**Figure 15 sensors-19-02394-f015:**
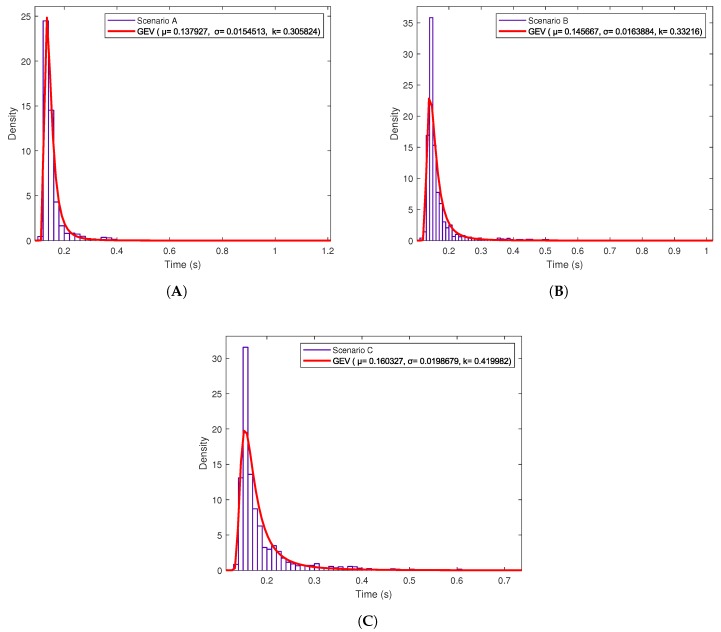
Probability Density Function (pdf) in OrbitDB for (**A**) Scenario A, (**B**) Scenario B and (**C**) Scenario C.

**Figure 16 sensors-19-02394-f016:**
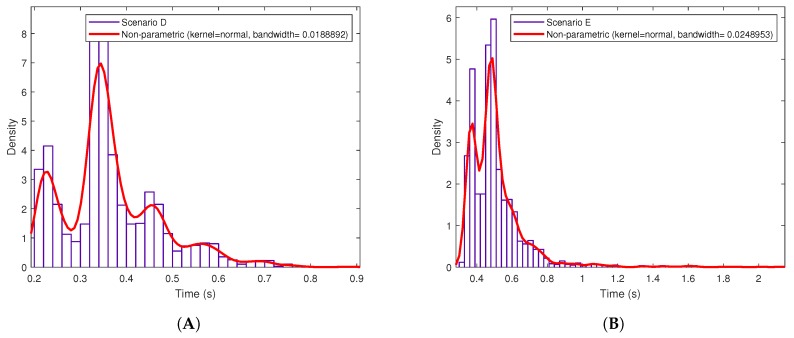
Probability Density Function (pdf) in OrbitDB for (**A**) Scenario D, (**B**) Scenario E and (**C**) Scenario F.

**Figure 17 sensors-19-02394-f017:**
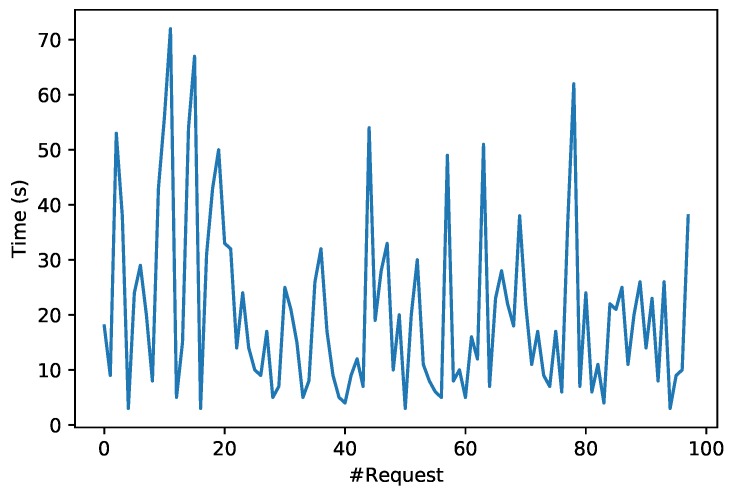
Ropsten testnet time response.

**Figure 18 sensors-19-02394-f018:**
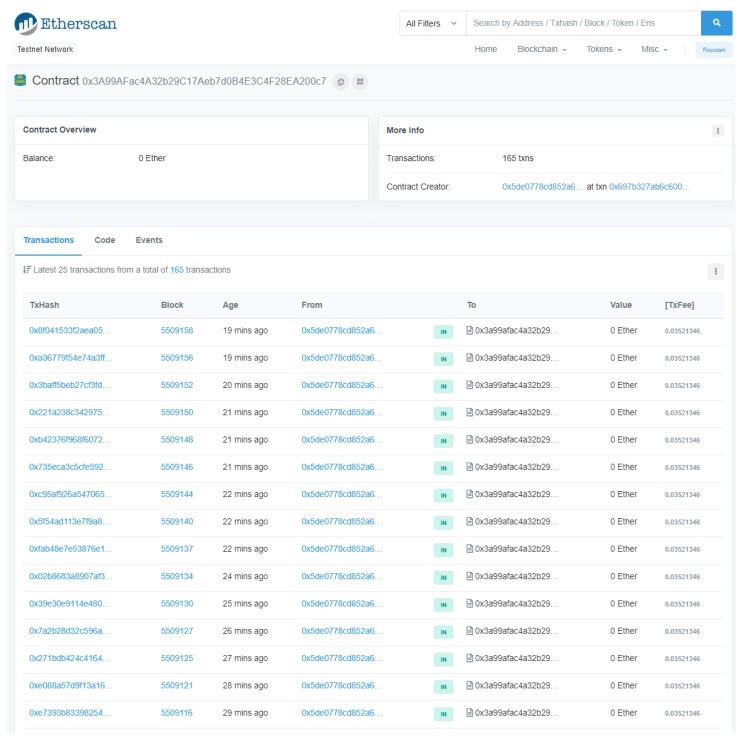
Caption of Ropsten Testnet transactions.

**Table 1 sensors-19-02394-t001:** Main characteristics of the most relevant communications and identification technologies for inventory and traceability applications.

Technology	Frequency Band	Max. Range in Optimal Conditions	Data Rate	Power type	Main Features	Main Limitations for Inventory Applications	Popular Applications
ANT+	2.4 GHz	30 m	20 kbit/s	Ultra-low power	Up to 65,533 nodes	Lack of commercial inventory tags	Health, sport monitoring
Barcode/QR	−	<4 m	−	No power	Very low cost, visual decoding	Need for LOS	Asset tracking and marketing
Bluetooth 5 LE	2.4 GHz	<400 m	1360 kbit/s	Low power	Batteries only last days to weeks	Batteries need to be recharged, shared communications radio frequency	Beacons, wireless headsets
DASH7/ISO 18000-7	315–915 MHz	<10 km	27.8 kbit/s	Very low power, alkaline batteries last months to years	Long reading distance, multi-year battery	Batteries need to be recharged, shared communications radio frequency	Smart industry and military
HF RFID	3–30 MHz (13.56 MHz)	a few meters	<640 kbit/s	No power	NLOS, no need for batteries	Relatively short reading range	Smart Industry, payments, asset tracking
Infrared (IrDA)	300 GHz to 430 THz	a few meters	2.4 kbit/s–1 Gbit/s	Low power	Low-cost hardware, security, high speed	Need for LOS, batteries may drain fast when transmitting continuously	Remote control, data transfer
IQRF	433 MHz, 868 MHz or 916 MHz	hundreds of meters	19.2 kbit/s	Low power	Long communications range	Shared communications radio frequency	Internet of Things and M2M applications
LF RFID	30–300 KHz (125 KHz)	<10 cm	<640 kbit/s	No power	NLOS, low cost	Very short reading distance (in general, a few centimeters)	Smart Industry and security access
NB-IoT	LTE in-band, guard-band	<35 km	<250 kbit/s	Low power	Long reading range	Dependent on third-party infrastructure	IoT applications
NFC	13.56 MHz	<20 cm	424 kbit/s	No power	Low cost	Short reading distance	Ticketing and payments
RuBee	131 KHz	20 m	8 kbit/s	Very low power	Magnetic propagation, multi-year battery life	Only one known manufacturer	Applications with harsh electromagnetic propagation
LoRa/LoRaWAN	2.4 GHz	kilometers	0.25–50 kbit/s	Low power	Long range, long battery life	Very few commercial inventory tags, more expensive than other alternatives	Smart cities, M2M applications
SigFox	868–902 MHz	50 km	100 kbit/s	Low power	Long range, global cellular network	Dependent on third-party infrastructure	Internet of Things and M2M applications
UHF RFID	30 MHz–3 GHz	tens of meters	<640 kbit/s	Very low power or no power	NLOS, wide range of suppliers, low cost	Propagation problems with metal and liquids (specially with high transmission frequencies)	Smart Industry, asset tracking and toll payment
Ultrasounds	>20 kHz (2–10 MHz)	<10 m	250 kbit/s	Low power	Based on sound wave propagation	Relatively short reading range	Asset positioning and location
UWB/IEEE 802.15.3a	3.1 to 10.6 GHz	< 10 m	>110 Mbit/s	Low power (batteries last hours to days)	Accurate positioning (centimeter accuracy)	Expensive hardware, propagation problems in metallic environments	Real Time Location Systems (RTLS), short-distance streaming
Wi-Fi (IEEE 802.11b/g/n/ac)	2.4–5 GHz	<150 m	up to 433 Mbit/s (one stream)	High power (batteries may last hours)	High speed, ubiquity	Short battery life	Internet access, broadband
Wi-Fi HaLow/IEEE 802.11ah	868-915 MHz	<1 km	>100 Kbit/s per channel	Low power	Long communications range	Not compatible with previous Wi-Fi standards, shared communications radio frequency	IoT applications
WirelessHART	2.4 GHz	<10 m	250 kbit/s	Low power (Batteries last several years)	Compatibility with HART protocol, standardized as IEC 62591	Shared communications radio frequency, lack of commercial inventory tags	Wireless sensor network applications
ZigBee	868–915 MHz, 2.4 GHz	<100 m	20–250 kbit/s	Very low power (batteries last months to years)	Easy to scale, up to 65,536 nodes	Relatively expensive hardware, potential interference from devices in the same frequency band	Smart Home and industrial applications

**Table 2 sensors-19-02394-t002:** Comparison of the main features of the most relevant UAV-based inventory systems and the proposed system.

Reference	Type of Solution	Labelling and Identification Technology	UAV Characteristics	Designed Architecture and Communications	Main Inventory Function	Experiments and Key Performance Indicators (KPIs)	Advanced Supply Management Data Techniques	Blockchain or Any Other DLT
[14]	Commercial solution by Hardis Group	Barcodes	Autonomous quadcopter with a high-performance scanning system and an HD camera. Battery life around 20 min (50 min to charge it).	It incorporates indoor localization technology. Automatic flight area and plan, 360${}^{\circ }$ anti-collision system.	Automate inventory-taking and inventory control in warehouses	No available KPI	Automatic acquisition of photo data. Cloud applications to manage mapping, data processing, reporting, and the fleet of drones. Compatible with all WMS and ERPS and managed by a tablet app.	No DLT
[15]	Commercial solution, Geodis and delta drone	Barcodes	Autonomous quadcopter equipped with four HD cameras	Indoor geolocation technology, it operates autonomously during the hours the site is closed.	Plug and play solution, this solution also adapts to all types of Warehouse Management Systems (WMS)	Reading rates close to 100%.	Enables the counting and reporting of data in real time, the processing of data, and its restitution in the warehouse’s information system.	No DLT
[16]	Commercial solution, Dron Scan	Barcodes	UAV equipped with a camera and a mounted display	DroneScan base station communicates via a dedicated RF frequency (not WiFi or Bluetooth) and has a range of over 100 m.	A Windows touch screen tablet allows the operator to receive live feedback both on screen and from audible cues as the drone scans and records data	50 times faster than manual capturing	All aspects of the imported data are customizable by modifying scripts, the customisation changes the way the system works and how the scanned data are processed. DroneScan software uploads scanned data and drone position information to the cloud (Azure IoT), to the customer systems (web services, RFC’s API’s or BAPI’s) and exports the data to Excel. The imported data are used to re-build a virtual map of the warehouse so that the location of the drone can be determined.	No DLT
[17]	Academic solution	RFID, multimodal tag detection.	Autonomous Micro Aerial Vehicles (MAVs), RFID reader and two high-resolution cameras	Fast fully autonomous navigation and control, including avoidance of static and dynamic obstacles in indoor and outdoor environments.	Robust self-localization solely based on an onboard LIDAR at high velocities (up to 7.8 m/s)	-	-	No DLT
[19]	Academic solution	-	-	-	Endogenous risk management mechanism to improve supply chain’s operation efficiency	Theoretical pharmaceutical factory supply chain topology structure based on blockchain.	Confront supply chain endogenous risk avoiding the credit risk caused by the information asymmetry among the enterprises inside the supply chain, and the risk caused by incomplete information acquisition inside the supply chain.	Blockchain and smart contracts
[23]	Academic solution	-	-	-	Autonomous economic system with UAV.	Although field trials were conducted with drones, no KPIs are available.	Architectural solution for organizing a business activity protocol for multi-agent systems.	Communication system between agents (DAOs) in a P2P network using Ethereum and smart contracts.
[59]	Academic solution	QR	IR-based camera, no additional description	Computer vision techniques (region candidate detection, feature extraction, and SVM classification) for barcode detection and recognition in factory warehouses.	Drone-assisted inventory management with an efficient detection framework to determine the localizations of 2D barcodes to improve path planning and reduce power consumption	Experiment performance results of 2D barcode images. The proposed method demonstrates a precision of 98.08% and a recall of 98.27% within a fused feature ROC curve.	-	No DLT
[60]	Academic solution, open-source code of the UWB hardware and MAC protocol software.	QR	-	Plug-and-play capabilities and minimal pre-existing infrastructure by combining two wireless technologies: sub-GHz for IoT-standardized long-range wireless communication backbone and UWB for localization.	A MAC protocol for an UWB localization system using battery-powered or energy harvesting operated anchors.	Experimental validation for two real-life scenarios: autonomous drone navigation in a warehouse mock-up and tracking of runners in sport halls.Theoretical evaluation of the design choices on overall system performance in terms of update rate, energy consumption, maximum communication range, localization accuracy and scalability.	-	No DLT
[61]	Academic solution	RFID	Phantom 2 vision DJI (weight 1242 g, maximum speed 15 m/s and up to 700 m)	Drone with a Windows CE 5.0 portable PDA (AT-880) that acts as a UHF RFID reader moves around an open storage yard.	Inventory checking in an open stock yard	Prototype. No performance experiments.	A data collection program detects and saves the information of passive tags obtained by a portable PDA. After the flight, the gathered tag data is transferred to the inventory checking server and is compared with the inventory data stored in database and classified in to four inventory states: normality, location error, missing, unregistered.	No DLT
[62]	Academic solution	RFID (EPC)	Draganfly commercial radio-controlled helicopters 82×82 cm, average flight time of 12 min	RFID readers attached to the simulated UAVs are assumed to have a 100% read-guarantee when EPC tags are within the reading range of the RFID reader.	Read the EPCs in the warehouse within the 12-min duration	Preliminary simulation results, three-dimensional graphical simulator framework has been designed using Microsoft XNA framework to represent a real warehouse	Coordinated distribution of the UAVs. Although six independent UAVs were deployed, they collectively failed to complete the task of finding all EPCs	No DLT
[63]	Academic solution	Barcodes, AR markers	UAVs and UGVs with LIDARs	UAV and UGV work cooperatively using vision techniques. The UGV acts as a carrying platform and as an AR ground reference for the indoor flight of the UAV. While the UAV is used as the mobile scanner.	Novel indoor warehouse inventory scheme to improve automation as well as the diminution of time consumption and injuries risks.	Experimental setup is to validate the visual guidance of the UAV taking the UGV as a ground. UAV need to be equipped with sensors to avoid collision with the racks during the scanning process.	-	No DLT
[67]	Academic solution	RFID	-	-	Physical Internet-based intelligent manufacturing shop floors	Experiments on logistics rules for optimizing the delivery time	Big data analytics framework that processes the information collected from an RFID-enabled shop floor	No DLT
Proposed system	Academic solution	RFID	Indoor/outdoor hexacopter designed from scratch as a trade-off between cost, modularity, payload capacity and robustness.	Modular and scalable UAV-architecture using WiFi infrastructure and ability to run decentralized applications.	Enable inventory and traceability applications focused on a holistic view at inventory levels across the supply chain and with external stakeholders	Prototype and performance experiments: inventory time in the warehouse under different circumstances, signal strength monitoring and performance of the implemented architecture (decentralized database and blockchain response latency)	Data distribution and enhanced cyber security (information integrity, tamper-proof data, ensured reliability and availability), efficient data storage and data versioning.	Decentralized database (OrbitDB) over InterPlanetary File System (IPFS) in a P2P network using Ethereum and smart contracts to automate certain processes.

**Table 3 sensors-19-02394-t003:** Main features of the UAV components.

Components	Relevant Features
Flight controllers	Pixhawk 2.4.8
	STM32F427 microcontroller
	STM32F103 coprocessor
Sensors	L3GD20 3-axis digital gyroscope
	LSM303D 3-axis accelerometer and magnetometer
	MPU6000 6-axis accelerometer and magnetometer
	MS5607 barometer
	GPS M8N
RFID reading system	NPR Active Track-2
	OrangePI PC Plus (SBC)
Additional components	Frame with six arms (550 mm of wingspan)
	Brushless motors 920 Kv
	ESCs Simonk 30 A
	Propellers: 10 inch-diameter and 45 inch-pitch
	Battery: 5 Ah (capacity) and 45 c-rate (discharge rate)

**Table 4 sensors-19-02394-t004:** Example of the collected inventory data.

# Read Tags	% Read Tags	Timestamp (HH:MM:SS,ms)	New Read Tag ID
0	0	18:14:43,087 (Take-off time)	
1	7.692307692	18:14:46,058	LOCATE00380349
2	15.38461538	18:14:46,090	RFCBDG00011185
3	23.07692308	18:14:48,558	LOCATE00380364
4	30.76923077	18:14:48,589	RFCBDG00011185
5	38.46153846	18:14:52,748	LOCATE00380372
6	46.15384615	18:14:54,349	LOCATE00380349
7	53.84615385	18:14:57,129	RFCBDG00011188
8	61.53846154	18:15:11,403	LOCATE00380330
9	69.23076923	18:15:33,008	LOCATE00365573
10	76.92307692	18:15:49,288	LOCATE00375358
11	84.61538462	18:15:56,454	LOCATE00380359
12	92.30769231	18:15:56,456	LOCATE00380357
13	100	18:16:01,630	LOCATE00375356

**Table 5 sensors-19-02394-t005:** Mean and variance of the use cases considered in the OrbitDB performance test.

# Tag IDs/Network	Intranet	Internet
13	Scenario A: μ: 0.1576 s; $\sigma^{2}$: 0.0039 s	Scenario D: μ: 0.3648 s; $\sigma^{2}$: 0.0114 s
5000	Scenario B: μ: 0.1669 s; $\sigma^{2}$: 0.0032 s	Scenario E: μ: 0.5063 s; $\sigma^{2}$: 0.0265 s
10,000	Scenario C: μ: 0.1892 s; $\sigma^{2}$:0.0049 s	Scenario F: μ: 0.5553 s; $\sigma^{2}$: 0.0093 s

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
