# Peer review of "Towards an Autonomous Industry 4.0 Warehouse: A UAV and Blockchain-Based System for Inventory and Traceability Applications in Big Data-Driven Supply Chain Management"

_sensors, 2019, doi:10.3390/s19102394_

Reviewer 1 Report

The manuscript presents a UAV-based system equipped with several sensors and a UHF RFID reader capable of improving the performances and the traceability levels of traditional industrialy inventory systems as well as of automating specific tasks. The entire system exploits a blockchain approach based on a distributed ledger in order to ensure data trustworthiness and validation, too.

POSITIVE ASPECTS

- The combination of UAVs, RFID-based approaches and DLTs (i.e., Blockchain) is worth of investigation and very interesting

IMPROVABLE ASPECTS

L25: I would suggest to use "paradigms and technological enablers" instead of "technologies"

L161-164: I disagree with the assertion "RFID evolved towards NFC" as these two technologies coexist and it should be more properly to differentiate between far-field and near-field communication (where the former group encompasses UHF RFID systems and the latter category coincides to NFC).

L204: reference 73 seems to me misplaced, as it should not be placed as the second reference in the big data section

Page 6, Table 1: I strongly suggest the authors to modify the table as it follows:

- split column "Power/Main Features" into two columns: "Power type" and "Main Features"

- add one additional column named "Main Limitations" 

- specify (for instance as a table footnote) that "Max range" values refer to optimal conditions

- max ranges proposed for BLE5 and NB-IoT are definitely over-estimated

L247: Blockchain approaches are not the only way to achieve decentralized applications: microservices-based architectures can achieve the same purpose

Figure 4: definitely too big: please shrink it a little

NEGATIVE ASPECTS

- The main concern I have is related to the usage of the blockchain approach: since several other applications for managing huge datastreams generated by RFID-based supply management activities are normally used nowadays (such as capturing applications based on complex events processors or even IoT-oriented cloud-based services from the most common public cloud providers), the authors should point out very clearly why their approach should be preferred to other, already existing, solutions. Moreover the Ethereum testnet scenario is briefly described at the end of Section 4 while more detailed explanations would be advisable.

- Second, the authors claimed in the abstract that a distributed ledger is used "to store certain inventory data collected by UAVs, validate them, ensure their trustworthiness and make them available to the interested parties". Although these usages are compatible with the proposed enabling technology, very few details are provided throughout the text. More specifically, the authors should clarify:

1) what data typologies are amenable to be managed in this way and why this approach can be considered as a better solution if compared to current alternatives for those data typologies

2) what blockchain-mediated data validation steps have been performed

3) what data trustworthiness assessments have been made

- Another negative aspect, specifically related to the management of the UAV configuration setup, pertains to the positioning of the reader's antennas on board: the authors should mention explicitly whether any consideration has been made in terms of potential distubing effects induced by the UAV's propellers on the reader's antenna performances. 

- Moreover, the considerations provided on SSI levels and their potential capability to help locating tags if an indoor GPS is available are too simplistic: several additional aspects (and issues) should be considered before claiming that the availabiility of SSIs can help to locate tags effectively (mutual effects between tags and surrouding materials, hostile EM environments, etc.)

Therefore, starting from such considerations, I would suggest tha authors to address these issues in a revised version of their manuscript.

Author Response

Dear Sir/Madam,

Please find below our detailed responses to the comments. In order to ease the labour of the reviewers we have colored in red the major differences with the previous version of the article.

Regards.

Reviewer 2 Report

This paper aims to present an autonomous industry 4.0 warehouse. Specifically, it develops a UAV and blockchain-based system for inventory and traceability applications in big data-driven supply chain management. My overall assessment is that the paper does not state a clear and specific problem to solve, and falls short in achieving the objective stated on the title of the manuscript. Most of the work reads as a review paper and for the design, methodology, and results, I do not consider there is a contribution.

The introduction presents the general motivation and scope of the paper consisting in Industry 4.0 and UAVs, while the second section presents a very extended literature review (the paper has 84 references). However, the main conclusion of the literature review is generic and the Authors have not been able to bring out the novel aspect of the work:

“it is still necessary more research to develop specific systems that consider the following main aspects: Data volume… speed… verification and veracity… versioning… accessibility”.

I suggest a shorter literature review but focused on a specific problem. Here the problem is generic. 

The design and implementation sections are rather short and they should be better connected with the strengths and weakness of the existing solutions of the specific problem the paper is trying to solve in order to highlight the contribution. Moreover, there are some decisions that must be clarified: the use of a hexacopter, why no other multirotor? the size of the UAV given that the payload is minimum (sensors and RFID reader); the use of a GPS as the experiments are indoors. The only thing that can be appreciated from these sections is the integration of existing technology, which is not a contribution itself.

Finally, the results only include four experiment runs for testing inventory time and signal strength. The results are clearly influenced by factors that are not taken into account in the methodology as the path of the UAV, the departing point, the pilot experience and skills, etc. The experiments must be designed to test the identified aspects presented in the literature review: data volume, speed, verification, veracity. The validation of the proposed approach must include more experiments and they must cover more conditions in order to generalize the findings. The results must include comparisons with other inventory systems based on UAVs such as the ones presented in [14-17] (just to mention some), not against a human operator that will be obviously outperformed.

Author Response

Dear Sir/Madam,

Please find attached our detailed responses to the comments. In order to ease the labour of the reviewers we have colored in red the major differences with the previous version of the article.

Regards.

Round  2

Reviewer 1 Report

All the comments provided in the first review have been properly addressed. Therefore, I propose to publish the manuscript in its actual form.

Reviewer 2 Report

I thank the authors for the effort made to improve their paper. Specifically, the inclusion of Section 5.4 for the performance evaluation of the architecture is acknowledged.

However, I still believe that the review of the literature is unnecessarily extensive. The literature review covers many topics and does not converge on a specific problem. Although the conclusions of the literature review increased (now in section 2.4), the problem remains broad and covers many aspects that should be addressed individually.

On top of this, a key aspect of my review is missing: the comparison of the results with a reported solution in the literature.

In general, what I see in this paper is the integration of existing technology more than a scientific contribution.